# Electrical resistance of individual defects at a topological insulator surface

Felix Lüpke[1,2], Markus Eschbach[2,3], Tristan Heider[2,3], Martin Lanius[2,4], Peter Schüffelgen[2,4], Daniel Rosenbach[2,4], Nils von den Driesch[2,4], Vasily Cherepanov[1,2], Gregor Mussler[2,4], Lukasz Plucinski[2,3], Detlev Grützmacher[2,4], Claus M. Schneider[2,3] & Bert Voigtländer[1,2]

Three-dimensional topological insulators host surface states with linear dispersion, which manifest as a Dirac cone. Nanoscale transport measurements provide direct access to the transport properties of the Dirac cone in real space and allow the detailed investigation of charge carrier scattering. Here we use scanning tunnelling potentiometry to analyse the resistance of different kinds of defects at the surface of a $(Bi_{0.53}Sb_{0.47})_2Te_3$ topological insulator thin film. We find the largest localized voltage drop to be located at domain boundaries in the topological insulator film, with a resistivity about four times higher than that of a step edge. Furthermore, we resolve resistivity dipoles located around nanoscale voids in the sample surface. The influence of such defects on the resistance of the topological surface state is analysed by means of a resistor network model. The effect resulting from the voids is found to be small compared with the other defects.

[1] Peter Grünberg Institut (PGI-3), Forschungszentrum Jülich, 52425 Jülich, Germany. [2] JARA-FIT, Forschungszentrum Jülich, 52425 Jülich, Germany. [3] Peter Grünberg Institut (PGI-6), Forschungszentrum Jülich, 52425 Jülich, Germany. [4] Peter Grünberg Institut (PGI-9), Forschungszentrum Jülich, 52425 Jülich, Germany. Correspondence and requests for materials should be addressed to B.V. (email: b.voigtlaender@fz-juelich.de).

I t has recently become clear that three-dimensional topological insulators (TIs) with their unique electronic properties are prime candidate materials for spintronic devices and quantum computing[1]. Prohibited 180° backscattering is ascribed to the topological surface states (TSSs) of these materials, which potentially leads to low power dissipation and high-efficiency devices[1,2]. Although research on these materials is still in an early stage, especially the compounds $Bi_2Se_3$, $Bi_2Te_3$ and $Sb_2Te_3$ have the potential to result in a disruptive technology due to their applicability at room temperature resulting from a pronounced band gap of up to 300 meV (refs 1,3).

Although strict 180° backscattering is prohibited in the TSS, it has become clear that phonon scattering and scattering at defects at different angles can significantly contribute to the resistance of the TSS[4,5]. Such backscattering at the surface of a TI has first been analysed by quasiparticle interference at steps and atomic defects[2,6–8]. Only recently, the electrical resistivity of single steps at the surface of $Bi_2Se_3$ was measured in locally resolved transport measurements[9]. However, when describing the nanoscopic resistance in a TI, steps are only one part of the whole picture and further defects such as domain boundaries and voids at the TI surface may have an important influence on the total sample resistance, as we will show in the following.

Here we use a combination of *in situ* surface analysis tools, angle-resolved photoemission spectroscopy (ARPES) and four-tip scanning tunnelling microscopy (STM), to achieve a detailed electronic and charge transport analysis of pristine $(Bi_{0.53}Sb_{0.47})_2Te_3$ thin films, respectively. For this compound, it has recently been shown that it has low bulk conductivity, whereas on its surface the Fermi energy cuts only through the TSS[10–13]. As a result, a current through the sample is expected to be transmitted predominantly by the TSS, which makes $(Bi_{0.53}Sb_{0.47})_2Te_3$ a promising candidate material for application in devices. Furthermore, it allows us to directly measure the nanoscopic resistance of individual defects, which we find at the TI surface. Consequently, we are able to determine the contribution of each type of nanoscale defects to the overall TI film resistivity. In particular, we find that domain boundaries result in the largest localized resistance, which is four times higher than that of a step edge. Furthermore, the resistance due to nanoscale voids in the TI surface is miniscule compared with these defects.

## Results

**Macroscopic sample characterization.** We investigate in the following a 10.5(5) nm film of $(Bi_{0.53}Sb_{0.47})_2Te_3$ prepared on a Si(111) substrate by molecular beam epitaxy[10,14]. We used the same sample for STM and ARPES measurements, which was kept under ultra high vacuum (UHV) conditions at all times to avoid any contamination of the TI surface throughout the measurements. The topography of the film measured by STM is shown in Fig. 1a. Here we find screw dislocations and step heights of 1 nm corresponding to quintuple layer (QL) steps as the dominant features in accordance with the literature[14]. Step heights different than 1 nm are only observed within a few nanometres away from the dislocation cores. From the average thickness of the TI film and the observed surface structure, we conclude that the TI film consists of 9 QL (the ninth layer is ~97% closed), with additional material distributed in form of islands in three open layers on top of the ninth QL. ARPES results of the surface obtained at $h\nu = 8.4$ eV are shown in Fig. 1b. Here, the TSS can be unambiguously identified as Dirac cone with the Dirac point located just above the valence band edge. By varying the photon energy to $h\nu = 21.2$ eV, we further find some contributions from the bulk conduction band minimum at the

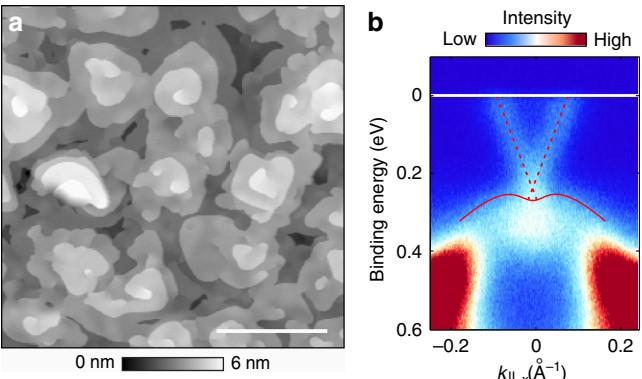

**Figure 1 | Overview of the sample topographic and electronic structure.** (**a**) STM image of the sample surface, showing that screw dislocations and QL steps are the dominant features. Scale bar, 200 nm. (**b**) ARPES measurement of the same sample along $k_{\parallel,x}$ direction (in-plane, corresponding to $\bar{K} - \bar{\Gamma} - \bar{K}$). The Fermi energy is indicated by the solid white line. The Dirac cone of the TSS is indicated by dashed red lines intersecting at the Dirac point just above the valence band edge (solid red curve).

Fermi energy due to its higher relative cross-section at this photon energy[15]. From this measurement, we determine the band gap at the TI surface to be 260(20) meV (see Supplementary Note 1 and Supplementary Fig. 1). The observed inherent *n*-doping of the TI surface we attribute to a downward band bending with respect to its bulk[16,17]. Spin-resolved energy distribution curves further reveal a high spin polarization of the Dirac cone, confirming the topological nature of the observed surface state (Supplementary Note 1 and Supplementary Fig. 1).

The macroscopic sample conductivity at room temperature is measured *in situ* in a four-probe geometry with a home-built four-tip STM[18]. Using equidistant tip spacing $s = 5–100$ µm, we find linear four-probe $I/V$ curves and constant four-probe conductance as a function of the tip spacing, as expected for a two-dimensional conductor[5]. The resulting sheet conductivity of the TI film is $\sigma_{4P} = 0.44(5)$ mS $\square^{-1}$. Furthermore, we determine the carrier concentration in the TSS from the ARPES measurements to be $n_{surface} = 4(1) \times 10^{12}$ cm$^{-2}$ (see Supplementary Note 1). A measurement of the electron mobility results in $\mu_{surface} \approx 500$ cm$^2$V$^{-1}$s$^{-1}$, which is in agreement with previously reported values in $(Bi_{1-x}Sb_x)_2Te_3$ with $x \sim 0.5$ (refs 11,12). From the above values we estimate the surface channel conductivity to be $\sigma_{surface} = en_{surface}\mu_{surface} = 0.32(11)$ mS $\square^{-1}$, which is also in agreement with recent transport measurements[9]. Compared with the total conductance of the entire TI film, this results in an estimated amount of $\sigma_{surface}/\sigma_{4P} \simeq 73\%$ of the total current through the TI film to be transmitted by the TSS channel at the sample surface. The remaining part of the current we expect to be transmitted by the TSS channel at the substrate interface along with possible bulk contributions[11,12,19].

**Nanoscale transport measurements.** In the following, we use scanning tunnelling potentiometry (STP) implemented into the four-tip STM, to determine the resistance of individual defects at the TI surface[20]. Hereby, a lateral current injected into the sample by two STM tips gives rise to a local current density $j$ and results in a voltage drop across the sample. A third STM tip is then scanned across the sample surface and simultaneously records topography and potential maps. Details on the present STP method can be found in ref. 20, Supplementary Note 2 and Supplementary Fig. 2. Figure 2a shows the result of a STP measurement at the TI surface. The dominant contributions to

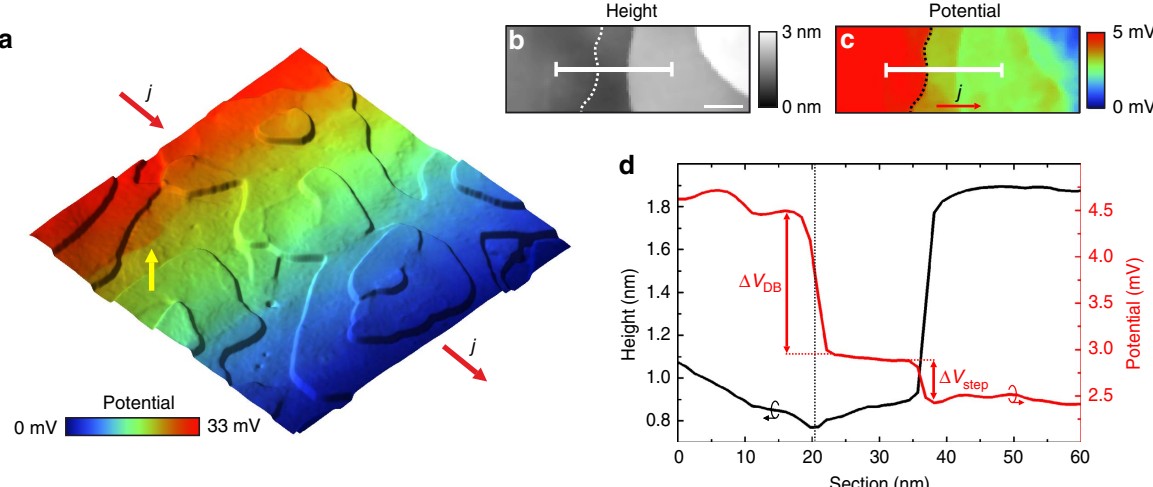

**Figure 2 | Nanoscale transport measurements at the sample surface.** (**a**) Overlay of topography as terrain and potential distribution as colour code. The current density $j$ through the sample is indicated by the red arrows. The topography is dominated by QL steps while in the potential we observe an overall linear voltage slope on the terraces and additional voltage jumps located along lines at the sample surface, for example, the one highlighted by the yellow arrow. Scan size: 300 nm. (**b**) Topography showing two steps at the sample surface. The section indicated by the solid white line is shown in **d**. Scale bar, 20 nm. (**c**) Corresponding potential map with subtracted linear background. Sharp voltage drops are located at the position of topographic steps and along the dotted line which we explain as a domain boundary in the TI film. The corresponding potential section indicated by the solid white line is shown in **d**. (**d**) Black line graph: height profile from **b**. Red line graph: potential section from **c**. The voltage drops are $\Delta V_{step} = 0.46(5)$ mV at the step edge and $\Delta V_{DB} = 1.54(5)$ mV at the position of the domain boundary (indicated by the vertical dotted line).

the voltage drop are an overall linear voltage slope on the terraces and voltage jumps along lines at the sample surface. A more detailed topography scan is shown in Fig. 2b, centred at a QL step. Figure 2c shows the corresponding potential map from which we subtracted the average slope on the terraces to reveal the defect induced fine structure in the voltage drop[9,20]. We observe a voltage drop at the position of the QL step but also an even more pronounced voltage drop at the indicated dotted line. In the corresponding height profile in Fig. 2d, we observe a minimum in the topography at the position of the dotted line, while the QL step can be identified by its height of $\sim 1$ nm. This finding at the position of the dotted line we explain to be due to a domain boundary between neighbouring domains of the TI film. The nucleation in islands, with two different rotational domains of the crystal structure[14], gives rise to in-plane stacking fault domain boundaries in the TI film upon coalescence of the islands[21]. In the corresponding potential section in Fig. 2d, the voltage drop located at the position of the domain boundary $\Delta V_{DB}$ is almost four times larger than that at the step edge $\Delta V_{step}$, which equals to a significantly lower conductivity of the domain boundary in comparison to the QL step edge.

To determine the absolute value of the conductivity of the defects, we use the above estimate that a fraction of 73% of the total current is transmitted by the surface channel: $I_{surface} = 73\% \cdot I_{total} = 653(11)$ µA with $I_{total}$ being the current injected by the outer tips. For a distance between the current injecting tips of $d = 10(2)$ µm, the local current density at the surface in the middle between the outer tips, where the potentiometry measurement is performed, is $j = 2I_{surface}/\pi d = 42(9) \text{Am}^{-1}$ (refs 20,22). The resulting conductivity of the two above line defects then is $\sigma_{step} = j/\Delta V_{step} = 907(250) \text{Scm}^{-1}$ and $\sigma_{DB} = j/\Delta V_{DB} = 272(60) \text{Scm}^{-1}$. Although the conductivity of the step is in agreement with the literature[9], the much lower conductivity of a domain boundary was not reported to this point. We explain this finding by the vertical extension of the domain boundary into the TI film beyond the first QL[21], in contrast to the step that is located exclusively at the first QL. Owing to the vertical extension of the TSS into the film[10], the

resulting scattering cross-section is larger for the domain boundary than for the step edge.

The terrace conductivity which we determine from the average voltage slope on the terraces $E_0$ is $\sigma_{terrace} = j/E_0 = 0.76(17) \text{ mS} \square^{-1}$. In total, we find that the line defects amount to 44% of the total observed surface channel resistance, in comparison with the voltage slope on the terraces, which contributes 56%. Hereby, the absolute voltage drops upon reversal of the lateral current are identical within the measurement errors (for further details, see Supplementary Note 3 and Supplementary Fig. 3). It is also noteworthy that possible bulk contributions to the transport would imply that defects have an even higher effect on pure TSS transport due to the resulting lower current density transmitted by the TSS.

**Transport dipoles around void defects.** Besides the above line defects, we further observe variations of the local electric potential in the form of resistivity dipoles around nanoscale voids at the sample surface. Resistivity dipoles result from a current flowing around a localized region of increased resistance[23,24]. Such an observation is of special interest in a TI due to the peculiar TSS properties, which prohibit 180° backscattering from such defects[4,5]. The topography of a typical void with a diameter of $\sim 5$ nm on a terrace of the TI film is shown in Fig. 3a. Figure 3b shows the corresponding potential map, where the potential slope of the surrounding terrace has been subtracted. The dipole is found to be centred at the defect and its lobes are aligned with the overall orientation of the current.

This observation qualitatively agrees with the classical analytic description of resistivity dipoles in diffusive transport, where the amplitude of the dipole depends only on the diameter of the void and the electric field on the surrounding terrace[23]. However, the analytic description considers only circular defects and deviations in shape can have significant impact on the observed dipole amplitude. For a detailed analysis of resistivity dipoles around arbitrarily shaped defects, we therefore use a resistor network model[25,26], to analyse the potential distribution around the voids (for details, see Supplementary Note 4 and Supplementary Fig. 4).

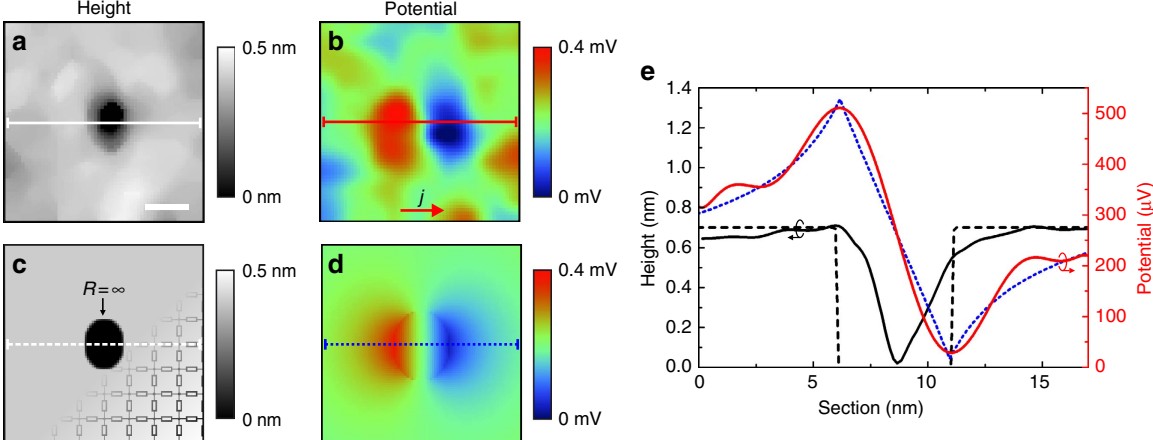

**Figure 3 | Resistivity dipoles around nanoscale voids.** (**a**) STM image of a typical void in the sample surface. Scale bar, 5 nm. (**b**) Corresponding potential map showing a dipole shaped feature centred at the defect. The lobes of the dipole are aligned with the macroscopic current direction. (**c**) Resistor network model mask with indicated schematic of the resistors. (**d**) Calculated potential distribution around the defect resulting from the resistor network model shown in **c**, after background subtraction. (**e**) Sections indicated in **a–d**. Solid black line: experimental height profile section from **a**. Solid red line: experimental potential section from **b**. Dashed black line: section of the model system shown in **c**. Dotted blue line: calculated potential section from **d**.

In this model too, only the dimensions of the void and the voltage slope on the terrace surrounding the defect, which is well defined in the experimental data, are parameters.

Figure 3c shows a plot of the resistor distribution mask corresponding to the experimentally observed defect geometry from Fig. 3a. Shown is a $80 \times 80$ nodal point excerpt from a total resistor network size of $200 \times 200$ nodal points. Hereby, the grey area corresponds to the conductivity of the surface channel, whereas black indicates an area of infinite resistance corresponding to a void in the sample surface. From this, the resulting potential distribution around the defect under static current flow is shown in Fig. 3d. For a comparison of theoretical and experimental results, Fig. 3e shows sections of the experimental (solid black line) and simulated (dashed black line) topography, and corresponding potential sections (red: experimental and blue dotted: simulated). In the experimental height profile, we find that the size of the void is smeared out laterally, which we attribute to the finite radius of the tunnelling tip. However, we observe larger voids to be 1 QL in depth, from which we conclude that the depth of the void shown in Fig. 3a is also 1 QL. The expected defect geometry is indicated by the dashed line. The section of the experimental potential map shows a pronounced dipole feature with a clear position of the dipole minimum and maximum. We find that the simulated potential distribution is in excellent agreement with the experimental data without any further parameters involved to fit the amplitude or shape of the defect. Furthermore, we conclude from the fact that we can explain the experimental data by means of a resistor network that we are in the diffusive transport regime[25,26]. In a next step, we will analyse the effect of voids on the overall resistance of the surface channel.

**Calculation of the void defect resistance.** In an infinitely wide conductor a single finite defect does not contribute to the resistance, because the relative area of increased resistance associated with the defect vanishes in comparison with the infinite conductor. Resistivity dipoles in this case can be described by the classical analytic description[23]. However, if a certain defect density is present, the conductor can be approximated by virtual parts (for example, squares for a two-dimensional conductor), each with a volume corresponding to the inverse of the defect density and with one defect residing in it (Fig. 4a). In each

constrained virtual part of the conductor, the single finite defect then results in an increased resistance compared with the defect-free conductor. This can be made clear in a simplified model shown in Fig. 4b where the conductor is modelled by a small number of parallel resistors and the defect in the conductor corresponds to a resistor of infinite resistance. Here, a current $j$ through the conductor has to be redistributed as it passed around the defect. At the position of the defect, this leads to a higher voltage drop due to the lower number of parallel resistors. The voltage drop amounts to $\Delta V = jR/2$ across the defect, whereas it is $\Delta V = jR/3$ elsewhere. The additional voltage drop caused by the defect results in a constant offset $\Delta V_{\text{defect}}$ in comparison with the undisturbed system and can be detected far away from the actual position of the defect. The increase in resistance resulting from the defect is then $\Delta R_{\text{defect}} = \Delta V_{\text{defect}}/j$. For the present example, this means that a current of 1A passing through such a virtual domain with resistors of $R = 1\,\Omega$ will result in a total voltage drop across the system of 1 V in the defect free case. With the defect replacing one of the resistors, the same current results in a voltage drop of $\sim 1.17$ V, which corresponds to an increase of the resistance due to the defect by 17%.

After introducing the effect of a defect on the resistance in the simplified, quasi one-dimensional model in Fig. 4b (no resistors in the vertical direction), we turn now to the full two-dimensional resistor network, which includes also resistors in the vertical direction as shown in Fig. 4c. This is the actual model we use to analyse the experimental data. Here, a void corresponds to a finite area with all resistors in this area having infinite resistance. Figure 4d shows the calculated potential resulting upon current flow through the model in Fig. 4c. A section through the centre of the dipole and parallel to the overall current direction is shown in Fig. 4e. Here, the minimum and maximum of the dipole feature are located at the boundary of the void (indicated as shaded area). Further away from the defect, the section approaches the indicated dashed lines corresponding to a persistent voltage drop of $\Delta V_{\text{defect}}$. Figure 4f shows sections perpendicular to the overall current direction. Here we find a bell-shaped potential distribution with decaying amplitude as the distance from the defect increases. It can be shown from the current conservation that the average value of each curve in Fig. 4f is the same. As a result, $\Delta V_{\text{defect}}$ is equal to the difference of the average voltage perpendicular to the overall current direction between any points along the $x$ direction before and after passing the defect.

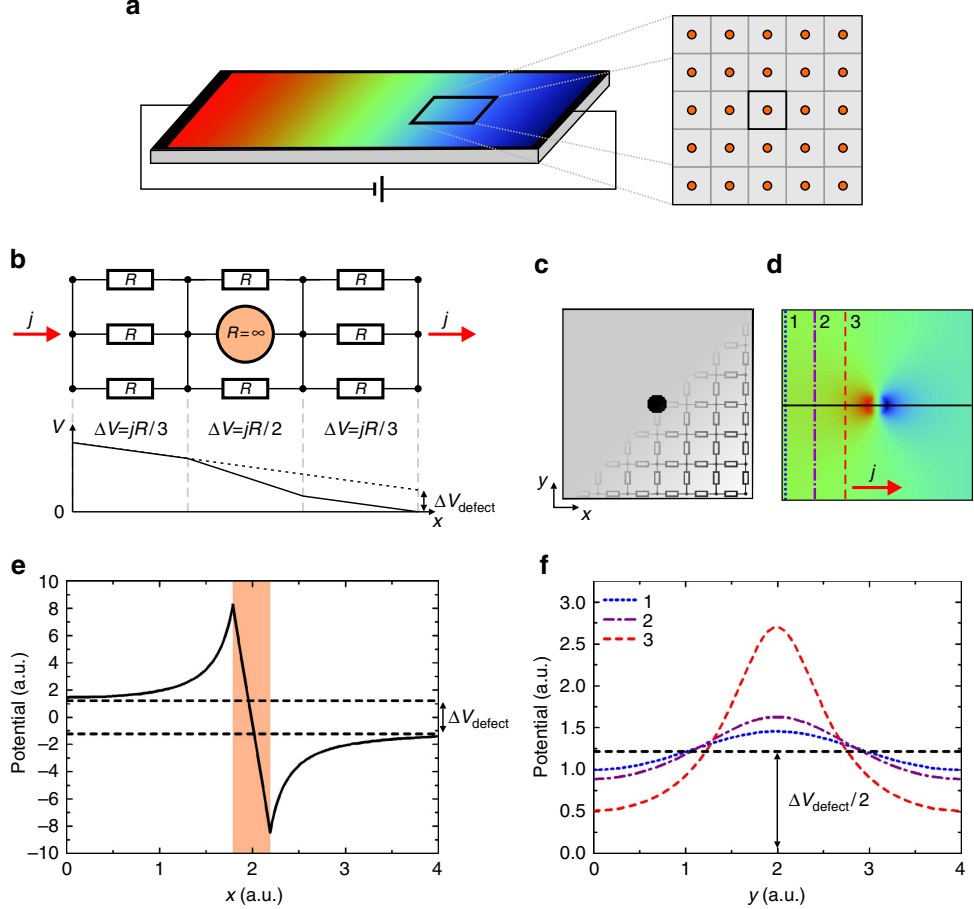

**Figure 4 | Resistor network analysis of the void defects.** (**a**) Schematic of a current carrying two-dimensional conductor. The conductor can be approximated by dividing it into virtual parts (squares) where each square corresponds to the inverse of the defect density in size and has one defect (orange circle) residing in it. (**b**) Simplified resistor network model of one virtual part of the conductor indicated in **a**. An incoming current $j$ is initially transmitted by the three parallel resistors left from the defect. However, at the position of the defect the current has to flow via smaller number of parallel resistors (two) before it can flow again via three resistors after it passed the defect. The resulting potential distribution shows a higher local voltage drop $\Delta V$ located at the position of the defect. The result is a voltage offset $\Delta V_{\text{defect}}$ compared with the voltage drop across a defect-free conductor, which results in a voltage drop corresponding to the dotted line. (**c**) Full resistor network model of a void in a conductor. Size: $200 \times 200$ pixel, with each pixel corresponding to a nodal point of the resistor network. (**d**) Background-subtracted potential distribution resulting from **c** upon current flow. (**e**) Section of the transport dipole along the solid black line in **d**. The position of the void is indicated by the shaded area. The persistent voltage drop $\Delta V_{\text{defect}}$ after the current passed the defect is indicated by the dashed lines. (**f**) Sections perpendicular to the current along the lines labelled as 1, 2 and 3 in **d** and with the corresponding line styles. The amplitude of the lateral potential distribution decays with increasing distance to the defect. The constant black dashed line corresponds to the average value of each section and equals to $\Delta V_{\text{defect}}/2$.

In the next step, we use the previous results to analyse the resistance arising from a density of voids at the TI surface corresponding to the experimentally found defect density and size. Therefore, we analyse a quadratic resistor network corresponding to one virtual domain of area $1/\rho$, where $\rho \approx 1/(150\,\text{nm})^2$ is the experimentally observed defect density, with a single defect of diameter $5\,\text{nm}$ in it. The resulting additional resistance associated with this defect density and size is $\Delta R_{\text{defect}} = \frac{\Delta V_{\text{defect}}}{j} = 2\,\Omega\square^{-1}$. In comparison, the experimental data results in $R_{\text{terrace}} = \frac{1}{\sigma_{\text{terrace}}} = 1315\,\Omega\square^{-1}$. This means that the experimentally observed void defect density and size results in an increase in the terrace resistance of only about 0.2% in comparison with the defect-free terrace.

## Discussion

Because of the small effect of the observed defect density and sizes on the sample resistance, we are not able to resolve $\Delta V_{\text{defect}}$ directly in the experiments. However, we show here that the knowledge of the geometrical arrangement of the defects (size and

defect density) is sufficient to calculate their influence on the resistance of the conductor. The above described increase in resistance due to a certain defect distribution becomes larger for increasing defect sizes and higher defect densities such that we expect that in a suitable sample this effect can be directly observed in the experimental data. Furthermore, this means that the terrace conductivity, which contributes to 56% of the total resistance of the surface channel, is dominated by phonon scattering and scattering at defects, which are smaller than the defects we report here, such as atomic vacancies or anti-site defects[5,13]. Owing to the influence of scattering at such small defects being still below our resolution, we only measure their influence on a larger scale, which manifests as an increased overall resistance on the terraces.

Tip jumping artefacts[27,28] can be excluded from playing a role in our STP measurements. The voltage drop features at steps are precisely located at their topographic signature and in agreement with the literature[9]. At domain boundaries, there is no significant change in the sample topography that could lead to tip jumping. Concerning the voids, the relevant signature of the resistivity

dipoles is the decay of the dipole outside of the actual void, namely on the surrounding flat terrace where also no significant change in the sample topography is present, which could lead to tip jumping. The influence of thermovoltage between tip and sample in the present measurements is insignificant: by measurement at zero transverse current we find the largest thermovoltage signal to be located directly at step edges where it is below $50\,\mu V$ in comparison with voltage drops of $\sim mV$ resulting from the transport measurements.

In conclusion, we determined the resistance of several types of nanoscale defects at the surface of a $(Bi_{0.53}Sb_{0.47})_2Te_3$ thin film. We find that significant amounts of resistance are present at surface step edges and domain boundaries of the TI film, which result in a fundamental limit of the resistivity of the TI film and therefore need to be carefully controlled for future application in electronic devices. By evaluation of voids in the sample surface, we further present a generic method how quasi 0D defects such as atomic vacancies and anti-sites defects can be evaluated in future local transport measurements. A way to further increase the resolution of the transport measurements, possibly enabling to resolve potential maps around single-atomic defects, would be the measurement at low temperatures[20]. Local transport measurements at atomic defects, below the carrier mean free path, should also enable to clarify quantum mechanical contributions to the electrical surface state transport.

## Methods

**Sample preparation.** Before the deposition by molecular beam epitaxy (MBE), a $10 \times 10\,mm^2$ Si(111) sample was cleaned by a RCA/hydrofluoric acid (HF) process to remove organic contaminations and the native oxide. The hydrofluoric acid passivates the Si surfaces with a hydrogen termination during the transfer into the MBE chamber (base pressure $1 \times 10^{-10}$ mbar). In the MBE chamber, the sample was first heated to $700\,°C$ for 10 min, to desorb the hydrogen termination from the Si surface and then cooled to $275\,°C$ for the TI film growth. The Te deposition was started for several seconds before the Bi and Sb evaporators were opened simultaneously. Co-evaporation of Te, Bi and Sb from effusion cells took place at $T_{Sb} = 408\,°C$, $T_{Bi} = 470\,°C$ and $T_{Te} = 330\,°C$. After deposition, the sample was cooled to room temperature and transferred to the four-tip STM and ARPES under UHV conditions for which a movable UHV chamber was used. For all results compared in the manuscript, the exact same sample was used. After the STM/ARPES measurements, the sample was characterized ex situ by X-ray reflectivity and Rutherford backscattering, to determine the TI film thickness and the chemical composition. Further details can be found in ref. 14.

**Angle-resolved photoemission spectroscopy.** ARPES measurements took place at $\sim 30\,K$ using a MBS A-1 hemispherical energy analyser, which was set to 40 meV energy resolution. We used monochromatic vacuum ultraviolet light with $hv = 8.4\,eV$ from a microwave-driven Xe discharge lamp and light with a photon energy of $hv = 21.2\,eV$ from a standard He discharge source.

**Four-probe measurement.** Four-probe measurements were performed at room temperature. To measure the macroscopic sample resistance, two of the four STM tips inject a lateral current through the TI layer, while the remaining two tips are used to measure the voltage drop at the surface resulting from the lateral current. Hereby, the tip positioning is monitored by a scanning electron microscope. Details on the home-built four-tip STM setup can be found in ref. 18. For all STM experiments, we used electrochemically etched tungsten tips.

**Scanning tunnelling potentiometry.** STP measurements were performed at room temperature. Hereby, a lateral current is injected into the sample using two STM tips, while the resulting voltage drop is mapped using a third tip in tunnelling contact. The technique was recently implemented into our four-tip STM as reported in ref. 20 and is based on the interrupted feedback technique. To map the sample topography and potential quasi-simultaneously, a software feedback loop alternatingly performs first topography and then potential measurement at each scan pixel. In detail, to measure the local sample potential, the tip is held at constant height while the tip voltage is adjusted by a feedback loop until the tunnelling current vanishes. At this voltage the electric potential of the tip is identical to that of the sample at the momentary scan position. The spatial and potential resolution are as low as Å and $\mu V$, respectively. Typical tunnelling parameters are $V_{tip} = -5\,mV$, $I_t = 10\,pA$. Further details on the potential measurement can be found in Supplementary Note 2 and Supplementary Fig. 2.

**Resistor network calculations.** The resistor network calculations solve Ohm's law $I = SV$ in the form of a linear equation system. Hereby, $S$ is the matrix of conductivities in which the resistance of each resistor enters inversely, $V$ is the vector of the voltage at each nodal point of the network and $I$ is the vector with the sums of incoming an outgoing currents at each nodal point. The latter, after Kirchhoff's law, is zero at each nodal point, except at the boundaries where current is injected. In the data shown here, these are the two boundaries left and right of the network, whereas across the boundaries at the top and bottom no current is allowed to flow in our out of the system. To solve the linear equation system, $S$ is inverted numerically $V = S^{-1}I$ such that $V$ contains the electric potential at each nodal point corresponding to the voltage drop across the system.

**Computer code availability.** Computer code used within the manuscript and its Supplementary Information are available from the corresponding author upon reasonable request.

**Data availability.** Data within the manuscript and its Supplementary Information are available from the corresponding author upon reasonable request.

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

### Acknowledgements

We gratefully acknowledge Helmut Stollwerk and Franz-Peter Coenen for technical assistance, and Helmut Soltner for fruitful discussions.

### Author contributions

F.L., M.E., T.H., M.L. and N.v.d.V. performed the experiments. F.L., V.C. and B.V. designed the STP experiment. M.E., T.H., L.P. and C.M.S. designed the photoemission experiment. M.L., P.S., D.R., N.v.d.V., G.M. and D.G. developed and fabricated the samples. The manuscript was written by F.L., M.E., T.H., M.L. and B.V. All authors discussed and commented on the manuscript.

### Additional information

**Competing interests:** The authors declare no competing financial interests.

**Publisher's note**: 

