## [Peer Review File · Nature Communications]

Reviewers' comments:

Reviewer #1 (Remarks to the Author):

The main motivation of this work is to find and report contribution of nanoscale defects to charge transport in a surface of 3D topological insulators (TIs). In this work, Dr. Bert Voiglander and coworkers report experimental measurement of resistances of various defects (domain boundary, voids, and quintuple layer steps). Using combination of scanning tunneling microscopy, potentiometry, and resistance network analysis, the authors' main finding is relatively large contribution of grain boundaries to the surface transport, while nanoscale voids contributes only less than a percent to the macroscopic surface resistance. They conclude the terrace conductivity is dominated by electron-phonon scattering or atomic level defects.

It is true that, although macroscopic surface dominated charge transport of TIs are routinely measured nowadays, nanoscopic transport are not well explored. The measurement of voltage drop along domain boundary is of the first with my best knowledge, and the data and numerical analysis is generally convincing. However, the importance or implications of these measurement does not fulfill criteria of Nature Communications and I recommend the authors to consider publishing more specialized journals in scanning microscopy.

1. I should point it out that the present measurement results does not provide any further insights about surface transport than just a observation of resistance of nanoscale defects. As TI surface state exhibit spin momentum locking, which is one of the main motivation for TI-based spintronic devices (and the authors actually mention spintronic devices as main application of TIs), any observation or analysis that can give insights to surface 'spin' transport can greatly improve the novelty and importance of the work. Along the same line, I feel the introduction section describing TI as nearly application-ready material is exaggerated compared to actual status of the field.
2. For the effect of void, the authors rely only on simple resistance network model to claim persistent voltage offset far away from nanoscale voids. This claim is not supported by experimental observation.
3. Another point (minor) is that, in the abstract section the authors points that nanoscale voids gives rise persistent voltage drop but does not mention that its net contribution to total resistance is minuscule as they conclude in the end. Without mentioning voids' final contribution, readers can get confused.

Reviewer #2 (Remarks to the Author):

Lüpke et al. report on nanoscale electrical resistances in thin films of a topological insulator ((BiSb)₂Te₃) which they analyze by scanning tunneling potentiometry. Thereby, the effect of local defects to the charge transport through the film is directly measured rather than by an indirect analysis of the electronic properties of the film. The authors identify surface step edges, domain boundaries in the film and nanoscale void defects to locally reduce the thin film's conductivity on a microscopic scale leading in sum to the macroscopic observed conductance/resistance of the film. Besides the already reported scattering of conduction electrons at surface step edges, the authors find that a large contribution to the film's resistance is also introduced by grain boundaries in the film. However, this is also expected for such defects since the domain boundaries represent two dimensional defects, affecting the whole film. Thus, the scattering cross section at grain boundaries is larger than for surface step edges and beside surface contributions also bulk contributions come into play. Moreover, at local voids the authors claim to observe local resistive dipoles which in sum give rise to a persistent voltage drop across the sample far away from this defect. These observations are further supported by a mathematical

model based on an Ohmic resistors network.

The paper is very well and clear written and deals with the “hot topic” on charge carrier transport in thin films of topological insulators (TIs) with nm scale lateral resolution. The data is of very high quality especially considering the difficult setup for such experiments. Although local resistances on TI thin films at step edges were observed before, here the experimental advance is that also grain boundaries and local voids are found to play a significant role for the transport properties in TI material. This information is valuable for researcher in physics analyzing TI materials but also for researchers in material science and engineering where TI film based devices shall be used and tuned for perspective future devices.

Thus, I believe that the audience of Nature Communications would benefit from the reported results which could further stimulate research and discussion in the affected fields. Therefore, I recommend publication of the manuscript in Nature Communications.

However, I have some minor comments/remarks the authors may want to consider prior to publication.

1. For me some of the figures appear too small, in particular fig 1a and fig 2a,b,c offer great detail on the surface quality and morphology which is difficult to see in the printed version of the manuscript. I recommend rescaling of the image in order to show all the details achieved by the measurements.

2. The pseudo three dimensional viewgraph in figure 2a looks nice but lacks some ease to identify topographic features and potential features. Maybe it would help if the spectator's angle of view would be more from above in order to keep the 3D topographic features together with the overlaid potential landscape.

3. In the manuscript it appears that the potentiometry measurements and the ARPES measurements were performed on the same sample and in the same UHV setup without any external transfer process of the sample. I rather expect that for both experiments new films were prepared and analyzed in the corresponding UHV chamber. Can the authors comment on this and make it clearer in the manuscript. This may be a somewhat critical issue, since the properties at the surface of a TI thin film heavily depend on the exact preparation conditions and/or contaminations.

4. On page 3 the authors report on screw dislocations and step edges with a step height of about 1nm. However, this somewhat contradicts to each other since at screw dislocations step heights between 0nm and 1nm should be found. Also this may affect the scattering at those step edges. Do the authors observe different voltage drops at these “different step heights? From the manuscript I understand that only the resistivity for the steps at 1nm height were determined.

5. What is the sample temperature during the STP measurements and does that anyhow affect the scattering, i.e. occurrence of voltage drops.

6. How does the voltage drop at the domain boundary scale as a function of the current density? From the corresponding slope one would extract the resistance/conductivity of an individual grain boundary. In particular it is important that at vanishing transverse current the voltage drop would also vanish in order to exclude artifacts from themovoltage effects. Also for both current directions the voltage drop should stay the same. Can the authoirs comment on this?

7. It would be nice to also show STP data for forward and backward direction e.g. in Fig. 2d to exclude any measuring artifact.

8. In the ARPES data (SI) the Fermi contour plot in Fig S1 looks rather asymmetric in intensity

(besides the warping). What is the reason for that?

9. In Figure S2 a color table scaling for the potential distribution is missing.

Reviewer #3 (Remarks to the Author):

The manuscript by Lukpe and co-authors reports on a study of epitaxial $(\text{Bi}_{0.53}\text{Sb}_{0.47})\text{Te}_3$ (BST), a topological insulator (TI), using local-probe potentiometry. The system utilized by the authors is very useful to probing surface currents: It comprises of an in-situ multi-probe measurement, where current is injected by the end probes and potential can be probed by tunneling measurements carried out using a middle probe. The integration of the measurement with STM allows the identification of transport features with topographical ones. Here the authors find two main line-defects, one associated with a step edge, and the other with a grain-boundary. In addition, they also identify point-defects where the potential assumes a dipole distribution (after background subtraction), a model which agrees very well with a simulation of a distributed resistor network.

This study is carefully executed, and the results are of sufficient importance to merit publication in Nature Communications. However, several points should be addressed, prior to publication.

1. The main issue relates to the leading contribution to resistance. Most of the 2nd part of the manuscript is related to the effect of one specific type of void – a single QL depression of few nm size. However, it appears that this void does not affect transport in any significant way. Given that there are many other types of defects (vacancies, anti-sites etc), which are often seen in STM measurements, it is not clear why these are not observed, and why aren't their specific contributions seen locally.

Other issues:

2. The technique of in-situ potentiometry is not as well known as STM. It is therefore important that an IV line-scan of the probe electrode be shown, and an explanation of how the potential is deduced from it given.

3. It is not clear how the sample is transferred from growth to STM / ARPES chambers – via “vacuum suitcase” or is the system multi-chamber. This can be noted in the methods section.

4. In Page 3: “The TI consists of 9 closed quintuple layers” – The authors should clarify what is a closed quintuple layer, and how is this number reached.

5. In the Methods section, the authors refer to “tip jumping”. They should clarify the meaning of this term.

6. It is somewhat surprising that even at the length-scale of few nm, transport is diffusive. I would have expected a ballistic length of few nm. Indeed, the potential variation is of order of 2-3 nm (Figure 3). This is attributed by the authors to be due to limitations of resolution. Could this be, in fact, a manifestation of ballistic length?

We would like to thank the reviewers for their constructive critique. Below we address the reviewers' comments point-by-point.

Reviewer #1 (Remarks to the Author):

1. I should point it out that the present measurement results does not provide any further insights about surface transport than just a observation of resistance of nanoscale defects. As TI surface state exhibit spin momentum locking, which is one of the main motivation for TI-based spintronic devices (and the authors actually mention spintronic devices as main application of TIs), any observation or analysis that can give insights to surface 'spin' transport can greatly improve the novelty and importance of the work.

We agree that the spatial resolution of spin scattering at quasi OD defects would be an ultimate goal for future experiments. For this purpose, we show here a general way how the scattering at quasi OD defects in topological insulators can be accessed experimentally and also analyzed. While In our experiments we are not able to resolve spin transport, we are confident that the present work will stimulate further experiments which should be able to resolve the spin transport at similar defects.

Along the same line, I feel the introduction section describing TI as nearly application-ready material is exaggerated compared to actual status of the field.

We agree with the reviewer and the introduction paragraph of the manuscript was changed accordingly (page 2, paragraph 2).

2. For the effect of void, the authors rely only on simple resistance network model to claim persistent voltage offset far away from nanoscale voids. This claim is not supported by experimental observation.

While it is difficult to resolve this effect directly in the present experiments due its small nature in the present sample, we present in the manuscript a generic concept how a resistance arises from nanoscale defects (page 12, paragraph 2). We are convinced that corresponding experimental studies will be performed in the near future.

3. Another point (minor) is that, in the abstract section the authors points that nanoscale voids gives rise persistent voltage drop but does not mention that its net contribution to total resistance is minuscule as they conclude in the end. Without mentioning voids' final contribution, readers can get confused.

We have changed the abstract on page 2, paragraph 1, to include the suggested information in order to make a clearer picture to the reader.

Reviewer #2 (Remarks to the Author):

1. For me some of the figures appear too small, in particular fig 1a and fig 2a,b,c offer great detail on the surface quality and morphology which is difficult to see in the printed version of the manuscript. I recommend rescaling of the image in order to show all the details achieved by the measurements.

We have increased the size of Figure 1 and 2 in the manuscript accordingly.

2. The pseudo three dimensional viewgraph in figure 2a looks nice but lacks some ease to identify topographic features and potential features. Maybe it would help if the spectator's angle of view

would be more from above in order to keep the 3D topographic features together with the overlaid potential landscape.

We have changed Figure 2 (a) as suggested by the reviewer for details to be better visible.

3. In the manuscript it appears that the potentiometry measurements and the ARPES measurements were performed on the same sample and in the same UHV setup without any external transfer process of the sample. I rather expect that for both experiments new films were prepared and analyzed in the corresponding UHV chamber. Can the authors comment on this and make it clearer in the manuscript. This may be a somewhat critical issue, since the properties at the surface of a TI thin film heavily depend on the exact preparation conditions and/or contaminations.

We have used the exact same sample for all measurement shown in the manuscript. The samples were transferred from the MBE chamber by a mobile UHV chamber, first to the STM chamber, then to the ARPES chamber. Hereby, the pressure was below $p < 1e-8$ mbar all the time and most of the time well below $p < 1e-9$ mbar. We have added this information to the methods section of the manuscript (page 3, paragraph 2 and page 16, paragraph 1).

4. On page 3 the authors report on screw dislocations and step edges with a step height of about 1nm. However, this somewhat contradicts to each other since at screw dislocations step heights between 0nm and 1nm should be found. Also this may affect the scattering at those step edges. Do the authors observe different voltage drops at these “different step heights? From the manuscript I understand that only the resistivity for the steps at 1nm height were determined.

At screw dislocations, step heights smaller than 1 nm are expected only very close to the dislocation core. Already few nm away from the dislocation core, the step height to the next terrace below corresponds to the quintuple layer height of 1 nm. We have added this information to the manuscript at page 3, paragraph 2.

5. What is the sample temperature during the STP measurements and does that anyhow affect the scattering, i.e. occurrence of voltage drops.

The potentiometry experiments were performed at room temperature. To point this out more clearly, we have added this to the methods section of the manuscript on page 16, paragraph 3. We expect that the charge carrier mean free path will be longer at low temperature. As a result, we expect the scattering, which we observe to be diffusive at room temperature, to shift towards the quantum mechanical limit for small defects such that it should be possible to identify contributions from ballistic transport. We have pointed out the implication of low temperature experiments more clearly in the conclusion, page 15, paragraph 2.

6. How does the voltage drop at the domain boundary scale as a function of the current density? From the corresponding slope one would extract the resistance/conductivity of an individual grain boundary. In particular it is important that at vanishing transverse current the voltage drop would also vanish in order to exclude artifacts from thermovoltage effects.

We have performed measurements at several different transverse currents, including measurements at zero transverse current in order to identify thermovoltage effects. From this we can say that thermovoltage effects in our measurements are small, not least because we made quite an effort in order to optimize the thermal stability during the measurements. At steps we observe a thermovoltage of less than 50 μV compared to the $\sim mV$ voltage drops observed in the transport experiments. From this we exclude artifacts as a result of

thermoelectric effects. We have added this information to the methods section of the manuscript on page 15, paragraph 1.

Also for both current directions the voltage drop should stay the same. Can the authors comment on this?

Concerning the polarity of the current, we have measured at the reverse currents and find no considerable variations of the absolute voltage drops as a function of the current polarity. (See also the next point)

7. It would be nice to also show STP data for forward and backward direction e.g. in Fig. 2d to exclude any measuring artifact.

We have added a corresponding section to the supplemental material (Supplementary Note 2 and Supplementary Fig. 2) showing reverse current directions plotted in the same graph to clarify the behavior upon current reversal for the interested reader. In the manuscript we have added a corresponding reference to the supplemental material on page 8 paragraph 1.

8. In the ARPES data (SI) the Fermi contour plot in Fig S1 looks rather asymmetric in intensity (besides the warping). What is the reason for that?

The asymmetry of the Fermi surface stems from the geometrical arrangement of the ARPES measurement setup. Hereby, the k_y direction is scanned by rotation of the sample with respect to the photon source. Due to the analyzer sitting in a fixed 45° angle with respect to the photon source, the resulting intensity in $-k_y$ direction is higher than in $+k_y$ direction. We have added this explanation to the Supporting information on page 2, paragraph 1.

9. In Figure S2 a color table scaling for the potential distribution is missing.

We have added the color scale bar to the graph in the supporting information (Supplementary Fig. 3).

Reviewer #3 (Remarks to the Author):

1. The main issue relates to the leading contribution to resistance. Most of the 2nd part of the manuscript is related to the effect of one specific type of void – a single QL depression of few nm size. However, it appears that this void does not affect transport in any significant way. Given that there are many other types of defects (vacancies, anti-sites etc), which are often seen in STM measurements, it is not clear why these are not observed, and why aren't their specific contributions seen locally.

The influence of scattering at defects on the single atom level (vacancies, anti-site defects) on the potential distribution is still below the resolution of our local transport measurements. As a result, we cannot resolve single transport dipoles at such small defects but only measure their influence on a larger scale which manifests as an increased overall resistance (linear voltage drop) on the terraces. This information was added to the manuscript on page 14, paragraph 2. By evaluation of void defects, we present a generic method how quasi 0D defects, including atomic vacancies and anti-sites defects can be evaluated in future local transport measurements. One way to increase the resolution of such transport measurements, possibly enabling to resolve the potential maps around such small defects would be the measurement at low temperatures. This outlook was added to the conclusion, page 15, paragraph 2.

2. The technique of in-situ potentiometry is not as well known as STM. It is therefore important that an IV line-scan of the probe electrode be shown, and an explanation of how the potential is deduced from it given.

We have added Supplementary Note 4 to the supporting information, which shows the I/V curve of the sample under investigation (Supplementary Fig. 4) and explains details of the working principle of the potential feedback with respect to the I/V curve.

3. It is not clear how the sample is transferred from growth to STM / ARPES chambers – via “vacuum suitcase” or is the system multi-chamber. This can be noted in the methods section.

The sample transfer took place via a movable UHV chamber. We have this information to the manuscript main text (page 3, paragraph 2) and methods section (page 16 paragraph 1).

4. In Page 3: “The TI consists of 9 closed quintuple layers” – The authors should clarify what is a closed quintuple layer, and how is this number reached.

We have modified the manuscript at page 3, paragraph 2, in order to clarify this point.

5. In the Methods section, the authors refer to “tip jumping”. They should clarify the meaning of this term.

Tip jumping is a common artifact in scanning tunneling potentiometry and details on this topic can be found in the literature. We have added references 27 and 28 in the manuscript, page 14, paragraph 3, where a detailed description of scanning tunneling potentiometry and related artifacts can be found.

6. It is somewhat surprising that even at the length-scale of few nm, transport is diffusive. I would have expected a ballistic length of few nm. Indeed, the potential variation is of order of 2-3 nm (Figure 3). This is attributed by the authors to be due to limitations of resolution. Could this be, in fact, a manifestation of ballistic length?

From the current experiments we are not able to determine the mean free path only that it has to be smaller than what we can describe by the use of a diffusive transport model. Further tailored measurements such as resolution of smaller defects or measurements at low temperature are required to further investigate on this topic. We have added this outlook to the conclusion paragraph, page 15, paragraph 2.

REVIEWERS' COMMENTS:

Reviewer #2 (Remarks to the Author):

The authors have addressed all the remarks and concerns from me and also from the other referees. The quality of the manuscript has further improved and I am certain that it convincingly fulfills the high quality standards of Nature Communications. The results are conclusive and will immediately attract the attention of the broad audience of Nature Communications so that I support the publication of this manuscript.

Reviewer #3 (Remarks to the Author):

Following the corrections made after by the Authors, I find this work can be accepted for publication.